# ResDCE-diff : Dynamic contrast enhanced MRI translation in prostate cancer using residual denoising diffusion models

Kishore Kumar[1]                                        KISHORE.M@HTIC.IITM.AC.IN
Sriprabha Ramanarayanan[1,2]                      SRIPRABHA.R@HTIC.IITM.AC.IN
Keerthi Ram[2]                                             KEERTHI@HTIC.IITM.AC.IN
Harsh Agarwal[3]                              HARSH.AGARWAL1@GEHEALTHCARE.COM
Ramesh venkatesan[3]                    RAMESH.VENKATESAN@GEHEALTHCARE.COM
Mohanasankar Sivaprakasam [1,2]                            MOHAN@EE.IITM.AC.IN

[1] *Department of Electrical Engineering, Indian Institute of Technology Madras (IITM), India*

[2] *Healthcare Technology Innovation Centre, IITM, India*

[3] *GE HealthCare, India*

**Editors:** Accepted for publication at MIDL 2026

## Abstract

Dynamic contrast enhanced MRI (DCE-MRI) identifies early perfusion patterns of aggressive prostate tumors, but its reliance on gadolinium contrast agents limits wider clinical adoption due to safety concerns. Recently, diffusion models offer a potential solution to synthesize contrast-enhanced images directly from non-contrast MRI. Previous diffusion models for prostate DCE-MRI require long inference times as they need hundreds or thousands of sampling steps limiting practical use. Moreover, the reverse generation process for DCE-MRI synthesis starts from pure noise without explicitly utilizing the prior information present in the non-contrast inputs in the diffusion process. We propose ResDCE-diff, a residual denoising diffusion model to synthesize early and late phase DCE-MRI images from non-contrast multi-modal inputs (T2-w, Apparent diffusion coefficient, and pre-contrast MRI). The diffusion process shifts anatomical, micro-structurally relevant and physics-informed residual features between the non-contrast inputs and DCE-MRI targets. Extensive experiments using PROSTATEx dataset show that ResDCE-diff, (i) consistently outperforms previous methods across early and late DCE-MRI phases with improvement margins of +1.29 db and +1.17 dB in PSNR, +0.04 and +0.03 in SSIM respectively, (ii) requires significantly lesser diffusion steps ($\approx 15$) compared to the baseline diffusion model, and (iii) exhibits relatively higher diagnostically relevant synthesis quality. The implementation is available at Github

**Keywords:** Prostate cancer, Dynamic contrast enhanced MRI (DCE-MRI), Diffusion model, Medical image-to-image translation

## 1. Introduction

Multi-parametric MRI provides a comprehensive assessment crucial for prostate cancer diagnosis and treatment guidance. The acquisition protocol integrates T2-Weighted (T2-w), Diffusion Weighted Imaging (DWI), Apparent Diffusion Coefficient (ADC), and Dynamic Contrast Enhanced MRI (DCE-MRI). The DCE-MRI uses gadolinium (Gad) based contrast agents to assess tumor angiogenesis and perfusion related to tumor aggressiveness (RM et al.,

2016). Based on PI-RADS scoring [1], DCE-MRI can highlight focal enhancements associated with clinically significant prostate cancer, minimizing unnecessary biopsies by about 25% (Armato et al., 2018) and over-diagnosis. However, gadolinium-based contrast agents raise safety concerns due to long-term deposition (BJ et al., 2018; M and S, 2016). Therefore, there is a need to minimize the dosage of Gad contrast or explore alternatives to Gad-based DCE-MRI. Contrast translation for MRI using generative deep learning methods have been

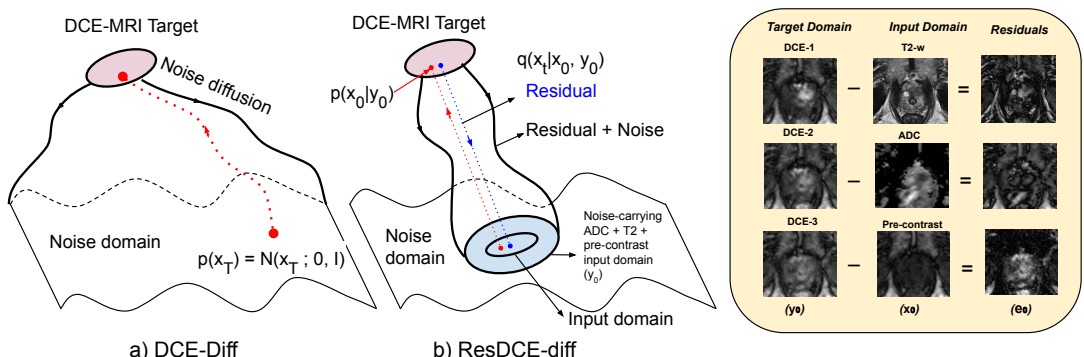

Figure 1: Concept diagram of DCE-diff (a) versus ResDCE-diff (b) adapted for DCE-MRI image translation. DCE-diff adopts noise diffusion, where the reverse process starts from pure noise. In contrast, ResDCE-diff employs residual shifting with a noise schedule, orienting the diffusion process from a prior non-contrast multi-modal data distribution. The multi-modal input domain is wider than the DCE-MRI target domain, emphasizing higher certainty in subject-specific perfusion information. (Right-side) residuals highlight structurally and physics-driven information explicitly utilized in ResDCE-diff for residual shifting. The above illustration is inspired by the diffusion framework presented in (Liu et al., 2024) and adapted to incorporate residual-shift diffusion for DCE-MRI synthesis.

explored including DCE-MRI synthesis for various anatomies like brain (Y et al., 2022b,a), breast imaging (RD et al., 2025; Zhang et al., 2023) and prostate imaging (Bharti et al., 2025), offering a potential alternative to gadolinium-based acquisitions (Y et al., 2025). These methods synthesize (i) late-response images from early-response images (CJ et al., 2021), (ii) a single or uni-modal MRI contrast images (Bone et al., 2021), and (iii) DCE-MRI sequences from multi-modal non-contrast MRI sequences (S et al., 2024), primarily utilizing Generative Adversarial Networks (GANs) as the framework for image generation.

Recently, Denoising diffusion probabilistic models (DDPM) (Ho et al., 2020) have have emerged as the state-of-the-art framework for image generation due to their strong generative capabilities and versatility (Yang et al., 2024).

DCE-diff (M et al., 2024) is based on DDPM for prostate non-contrast to DCE-MRI translation. DCE-diff incorporates diffusion of DCE-MRI images by infusing information

---

1. https://www.acr.org/Clinical-Resources/Clinical-Tools-and-Reference/Reporting-and-Data-Systems/PI-RADS

from multi-modal non-contrast MRI images as conditional inputs. However, the reverse process in DCE-Diff starts from Gaussian noise, requiring a large number of steps, making the process inefficient during inference as shown in Figure 1(a). Moreover, the prior information about the tumor present in the ADC, T2-w and pre-contrast MRI is only used as a conditional input to the network. This implicit usage of non-contrast information in the diffusion process undermines the synthesis of anomalous regions in DCE-MRI output.

Residual denoising diffusion (Liu et al., 2024) models can direct the diffusion process from the target image to the conditional input image as shown in Figure 1(b) by shifting the residual between them step by step. The noise scheduler in ResShift (Yue et al., 2023) controls the shifting speed of the residual and the noise strength in each step, substantially improving the transition efficiency for super-resolution tasks. Unlike super-resolution, whose residuals captures only image degradation, multi-modal contrast translation integrates multiple modality specific signals making the generative mapping inherently many-to-many and substantially more complex.

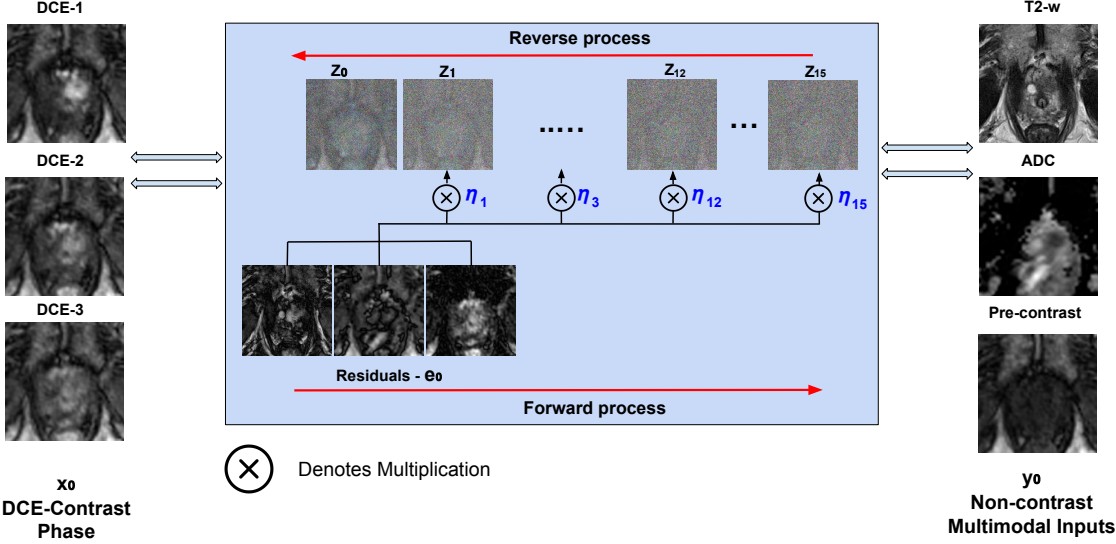

Figure 2: Overview of the proposed residual-shift diffusion framework for synthesizing DCE-MRI contrast phases $(x_0)$ from conditional non-contrast multi-modal inputs $(y_0)$. Residual maps $e_0$ between the target DCE phase $x_0$ and multi-modal inputs $y_0$ with a shifting sequence $\eta_t$ are progressively noised over 15 diffusion steps. The reverse diffusion process iteratively denoises to reconstruct the DCE- onset, DCE Early-phase and DCE Late-phase from its non-contrast inputs.

We consider the problem of translating non-contrast prostate MRI images to DCE-MRI onset, early and late phase responses using residual denoising diffusion process as illustrated in Figure 2. Specifically, the residual shifting incorporates multifaceted features that differentiate non-contrast inputs (T2-w, ADC, and pre-contrast MRI) from DCE-MRI early and late response sequences. Our method explores diffusion process using residual in

three dimensions, (i) anatomical residual features between T2-w MRI and onset time point or DCE-1 before early phase contrast uptake,

(ii) micro-structurally relevant residual features between ADC and early contrast-uptake time point or DCE-2 (iii) physics-informed residual features between pre-contrast MRI and late phase contrast uptake time point or DCE-3 to encode maximum contrast enhancement before leakage.

Our approach explicitly utilizes the non-contrast information in the diffusion process, unifying synthesis task that requires subject-specific certainty or diversity (generation of perfusion patterns) and subject-agnostic information (anatomy - prostate and bladder). Our contributions are,

- We propose ResDCE-diff, a residual denoising diffusion model to synthesize early and late-phase prostate DCE-MRI images from multi-modal non-contrast inputs such as T2-w, ADC, and T1 pre-contrast images.

- We extend a residual shifting diffusion model beyond super-resolution to the task of multi-modal DCE-MRI translation by redefining the residual between non-contrast and DCE-MRI. The residuals capture the distinctive and correlated intra-modality differences and explicitly guide the reverse generation process for DCE-MRI synthesis from non-contrast sequences.

- We quantitatively and qualitatively validate the method against GAN and diffusion baselines, reporting improvement margins of +1.29 dB and +1.17 in PSNR, +0.04 and +0.03 in SSIM for early- and late- phase synthesis.

## 2. Related works

Deep learning-based synthesis of contrast-enhanced MRI has emerged as a promising alternative to gadolinium-administered imaging, with deep learning methods have been extensively explored in MRI for segmentation (Kompella et al., 2019) and reconstruction (Beauferris et al., 2022; Ramanarayanan and Murugesan, 2020; Ramanarayanan et al., 2023, 2025) and early work focusing on generating post-contrast images from pre-contrast scans in breast and brain to preserve lesion hyper-intensities and tumor assessment (Y et al., 2022b; Zhang et al., 2023; RD et al., 2025). With this, several GAN-based methods have been proposed: TSGAN uses tumour segmentation masks and a lesion-focused discriminator to guide post-contrast breast MRI synthesis, while AAD-DCE leverages global and local attention in the discriminator to better exploit multi-modal prostate inputs for early- and late-phase DCE synthesis (Kim et al., 2023; Bharti et al., 2025; Murugesan et al., 2019). RegGAN addresses misalignment in paired medical image translation by coupling a registration network with the GAN and treating misregistered labels as noisy targets, and ResViT introduces a hybrid CNN–transformer generator to capture long-range multi-modal context in MRI synthesis (Kong et al., 2021; Dalmaz et al., 2022). More recently, DCE-diff replaced GANs with a conditional DDPM for prostate DCE-MRI translation, achieving improved image quality and cross-scanner robustness, but it still requires hundreds to thousands of denoising steps from pure noise, leading to substantial training and inference times (M et al., 2024). Recently, MRI-to-multi-modal PET synthesis has explored using prior-guided residual dif-

fusion by optimizing inter-domain (MRI-PET) residual losses, but cross-modality residuals are inherently less correlated than intra-modality contrast differences (Ou et al., 2024).

Our work concurs with this work in residual shifting, with focus on intra-modality differences, mapping multi-modal non-contrast inputs to early- and late- phase DCE sequences.

## 3. Methodology

The overview of our proposed method is shown in Figure 2. The residual shifting mechanism of ResDCE-diff builds up a Markov chain to facilitate the transition from non-contrast MRI modalities into their corresponding post-contrast DCE-MRI representations. The diffusion process involves forward and reverse process. Let $\mathbf{y}_0 = (y_0^{\text{T2}-\text{w}}, y_0^{\text{ADC}}, y_0^{\text{Pre}-\text{contrast}}) \in \mathbb{R}^{H \times W \times 3}$ denote the multi-modal non-contrast input composed of T2-w, ADC, and pre-contrast images and $\mathbf{x}_0 = (x_0^{\text{Onset}}, x_0^{\text{Early}}, x_0^{\text{Late}}) \in \mathbb{R}^{H \times W \times 3}$ denote the target post-contrast images, containing three contrast-enhanced phases (onset, early-phase and late-phase). The post-contrast MRI exhibit a strong correlation with the non-contrast inputs for a given individual. Due to the contrast uptake pattern, the ADC and pre-contrast can substantially contribute to the residual features.

### 3.1. Forward process

In the forward process, the goal is to gradually shift the DCE-MRI image $x_0$, towards the non-contrast multi-modal inputs $y_0$. Let $e_0 = y_0 - x_0$ be the residual difference between $x_0$ and $y_0$, which represents how the contrast appearance (DCE-1, DCE-2, DCE-3) deviates from its non-contrast counterpart (T2-w, ADC, pre-contrast). To achieve this, a monotonically increasing shifting sequence $\{\eta_t\}_{t=1}^T$ with timestep $t$ satisfies $\eta_1 \to 0$ and $\eta_T \to 1$ determines the fraction of residual addition at each step t. At each forward diffusion step $t$, the forward transition distribution based on the shifting sequence is defined as:

$$q(x_t \mid x_{t-1}, y_0) = \mathcal{N}\left(x_t;\, x_{t-1} + \alpha_t \cdot e_0,\, \kappa^2 \alpha_t I\right), \tag{1}$$

where $\alpha_t = \eta_t - \eta_{t-1}$ and $\kappa$ is a hyperparameter controlling variance of noise and I is the identity matrix. At any timestep $t$, the marginal distribution of $x_t$ given $(x_0, y_0)$ can be written as:

$$q(x_t \mid x_0, y_0) = \mathcal{N}\left(x_t;\, x_0 + \eta_t \cdot e_0,\, \kappa^2 \eta_t I\right), \tag{2}$$

Therefore, $x_t$ can be re-parameterized as:

$$x_t = x_0 + \eta_t e_0 + \kappa \sqrt{\eta_t}\, \epsilon, \qquad \epsilon \sim \mathcal{N}(0, I), \tag{3}$$

The above construction (Equation 2) induces a marginal probability path at time $t$, from the DCE-post contrast distribution $\delta_{x_0}(\cdot)$ for each data point $x_0$ towards the multi-modal input distribution $\mathcal{N}(\cdot\,; y_0, \kappa^2 I)$ at time $T$.

### 3.2. Reverse process

In the reverse process, the noisy non-contrast inputs $(y_0)$ is progressively restored into DCE-MRI outputs $(x_0)$ by estimating the posterior distribution $p(x_0 \mid y_0)$ as:

$$p(x_0 \mid y_0) = \int p(x_T \mid y_0) \prod_{t=1}^{T} p_\theta(x_{t-1} \mid x_t, y_0) \, dx_{1:T}, \tag{4}$$

The parameterized reverse transition kernel $p_\theta(x_{t-1} \mid x_t, y_0)$ approximates the true reverse distribution from $x_t$ to $x_{t-1}$ given the current state $x_t$ and conditional non-contrast $y_0$. We therefore adopt commonly used diffusion model (Dhariwal and Nichol, 2021; Song et al., 2020) assumptions for $p_\theta(x_{t-1} \mid x_t, y_0) = \mathcal{N}(x_{t-1}; \mu_\theta(x_t, y_0, t), \Sigma_\theta(x_t, y_0, t))$. The fixed model variance independent of $x_t$ and $y_0$ and the re-parameterized mean are given as:

$$\Sigma_\theta(x_t, t, y_0) = \kappa^2 \frac{\eta_{t-1}}{\eta_t} \alpha_t I, \tag{5}$$

$$\mu_\theta(x_t, t, y_0) = \frac{\eta_{t-1}}{\eta_t} x_t + \frac{\alpha_t}{\eta_t} f_\theta(x_t, t, y_0), \tag{6}$$

where $f_\theta(\cdot)$ is a neural network with learnable parameters $\theta$ that predicts $x_0$ from $x_t$, $y_0$ (non-contrast inputs) with the objective function defined below as:

$$\min_\theta \ \mathbb{E}\left[\ \left\| f_\theta(x_t, y_0, t) - x_0 \right\|_2^2 \right]. \tag{7}$$

## 4. Experimental Setup

### 4.1. Dataset Description and Evaluation Metrics

We utilized a publicly available PROSTATEx (Litjens and et al., 2017) dataset, which contains multi-parametric MRI data from 346 studies acquired without an endo-rectal coil. Each study comprises of T2-w, ADC, DCE pre-contrast, and post-contrast DCE sequences. We split the data into train, validation, and test sets in an 80:10:10 ratio. All volumes were rigidly registered using SimpleITK, and the prostate region was center-cropped to a size of $160 \times 160 \times 16$. In addition, we evaluated our method on the prostate-MRI dataset (Choyke et al., 2016) which includes 26 patients comprising of T2-w, diffusion-weighted imaging(DWI) pre-contrast T1-weighted and post contrast DCE-MRI acquisitions. ADC maps were derived from the DWI images. To ensure consistency with PROSTATEx identical preprocessing steps including rigid registration using SimpleITK and center cropping to 160 X 160 x 16 were applied. We assess and compare the translation performance using PSNR (Peak Signal-to-Noise Ratio), SSIM (Structural Similarity Index Measure), and MAE (Mean Absolute Error).

### 4.2. Implementation Details

We implemented all baseline methods and our proposed model using PyTorch (v1.12), with experiments conducted on a single NVIDIA RTX 3090 GPU (24 GB memory). Input images, including both target and conditional modalities, were uniformly processed at a spatial resolution of $160 \times 160 \times 3$. Training was performed using the Adam optimizer with default PyTorch parameters, a mini-batch size of 8, and a constant learning rate of $5 \times 10^{-5}$ over 50,000 iterations. We adopted the noise scheduling approach proposed in ResShift (Yue et al., 2023) to govern the diffusion process.

The reverse process architecture is based on a U-Net backbone enhanced with Swin-Transformer blocks replacing conventional self-attention layers, enabling improved modeling of spatial relationships and cross-channel interactions in multi-modal settings. The forward process hyper parameters were configured as $\kappa = 2.0$ for noise scaling and $p = 0.3$ for distribution shifting. We used $T = 15$ diffusion time steps for both training and inference phases.

## 5. Results and Discussion

### 5.1. Quantitative and Qualitative results

Table 1 presents the quantitative comparison of ResDCE-diff method compared with the baselines, including several CNN-based generators such as Pix2Pix (Isola et al., 2017), Reg-GAN (Kong et al., 2021), TSGAN (Kim et al., 2023), and Transformer-based generators such as MINet (Feng et al., 2021a), Task-Transformer Net (Feng et al., 2021b), ResViT (Dalmaz et al., 2022), DCE-former (S et al., 2024), and diffusion based methods such as DCE-diff (M et al., 2024). Our method achieves a PSNR gain of +1.29 dB and an SSIM increase of +0.04 over DCE-diff (requiring 1000 steps) in the early phase, and +1.17 dB PSNR and +0.03 SSIM in the late phase, while maintaining a comparable MAE. Additionally, for the onset phase, our method attains a PSNR of $23.766 \pm 1.77$, an SSIM of $0.72 \pm 0.06$, and an MAE of $0.04 \pm 0.01$. For consistency with the baseline methods, onset-phase metrics are not shown in Table 1

Table 1: Quantitative comparison of generated early- and late- response DCE-MRI images between state-of-the art models.

| MODEL | EARLY-RESPONSE | | | LATE-RESPONSE | | |
|---|---|---|---|---|---|---|
| | PSNR↑ | SSIM↑ | MAE↓ | PSNR↑ | SSIM↑ | MAE↓ |
| Pix2Pix | $15.21 \pm 5.49$ | $0.28 \pm 0.18$ | 0.11 | $15.29 \pm 1.36$ | $0.25 \pm 0.07$ | 0.12 |
| RegGAN | $20.56 \pm 0.02$ | $0.59 \pm 0.02$ | 0.05 | $20.09 \pm 0.02$ | $0.58 \pm 0.02$ | 0.06 |
| TSGAN | $21.16 \pm 3.50$ | $0.62 \pm 0.10$ | 0.06 | $20.46 \pm 2.64$ | $0.59 \pm 0.09$ | 0.07 |
| ResViT | $21.46 \pm 0.04$ | $0.63 \pm 0.04$ | 0.06 | $20.88 \pm 0.04$ | $0.62 \pm 0.05$ | 0.06 |
| MINet | $22.01 \pm 2.76$ | $0.66 \pm 0.15$ | 0.07 | $21.65 \pm 3.06$ | $0.63 \pm 0.19$ | 0.08 |
| Task Transformer (T2Net) | $22.13 \pm 3.20$ | $0.65 \pm 0.17$ | 0.06 | $21.37 \pm 2.48$ | $0.64 \pm 0.16$ | 0.07 |
| DCE-diff (1000 steps) | $22.10 \pm 1.79$ | $0.67 \pm 0.05$ | **0.04** | $21.73 \pm 1.95$ | $0.65 \pm 0.06$ | 0.05 |
| DCE-former | $22.85 \pm 1.71$ | $0.70 \pm 0.05$ | 0.05 | $22.07 \pm 1.93$ | $0.67 \pm 0.06$ | 0.06 |
| ResDCE-diff (15 steps) | $\mathbf{23.39 \pm 1.58}$ | $\mathbf{0.71 \pm 0.08}$ | 0.05 | $\mathbf{22.90 \pm 1.87}$ | $\mathbf{0.68 \pm 0.08}$ | **0.05** |

The visual comparison in Figure 3 further supports the numerical findings by producing sharper, perfusion-related enhancements learned through anatomical and physics-informed residual images within the diffusion process. The residual error maps confirm that our predictions adhere more closely to the ground truth compared with DCE-diff and other CNN and transformer based baseline methods. This performance gain is attributed to the residual shifting formulation into the diffusion process, where the model learns the residual component between multi-modal non-contrast inputs and DCE-MRI sequences in addition to the conventional noise component. Through this embedding of residual into the forward noising process, the transitional kernel guides the target distribution towards its corresponding pre-contrast counterpart in an oriented manner. As a result, the process

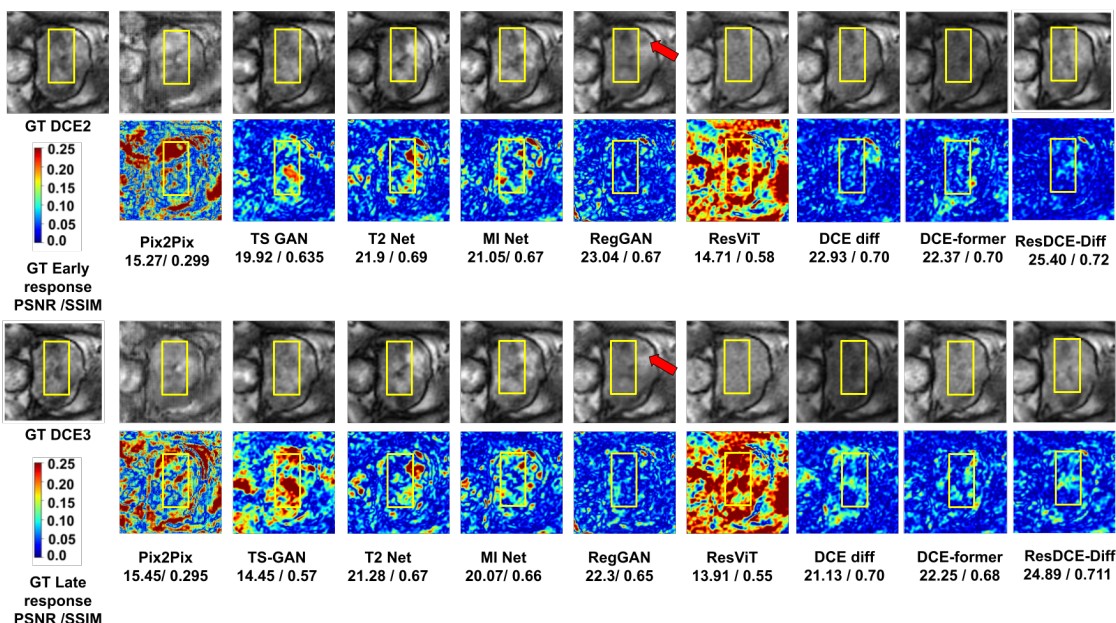

Figure 3: Qualitative results of ResDCE-diff and other state-of-the-art models trained and tested on PROSTATEx dataset. The regions marked on yellow highlights hypo-intense regions within the transverse zone of the prostate. The ResDCE-diff shows plausible structure with lesser residual errors compared with other methods. Although RegGAN gives comparable residual error, the structure are smooth. The hyper-intense regions surrounding the region of interest, as indicated by a red arrow shows artifacts. The proposed design starts with non-contrast MRI data distribution that correlates with the corresponding paired DCE data with subtle changes in regions with perfusion, as against starting with pure noise in the case of DCE-Diff.

enables effective synthesis of contrast-enhanced images even with fewer sampling steps (15 steps) when compared with DCE-diff (requires 1000 steps), leading to faster training and inference without compromising synthesis quality. Our methods achieves a 3.56x faster inference speed comparatively while maintaining the synthesis quality per sample. Overall, these findings shows the computational efficiency and effectiveness of embedding physiologically meaningful residuals within the diffusion process for DCE-MRI synthesis. In addition to synthesis, ResDCE-diff also requires significantly fewer parameters and provides faster sampling as shown in Table 2 and Table 3 than DCE-diff, yet delivers consistently better synthesis.

Table 2: Model size comparison in terms of number of parameters (in millions).

| Model | Pix2Pix | RegGAN | ResViT | DCE-Former | DCE-diff | ResDCE-diff |
|---|---|---|---|---|---|---|
| Parameters (M) | 11.40 | 11.38 | 127.82 | 20.42 | 125.08 | 58.38 |

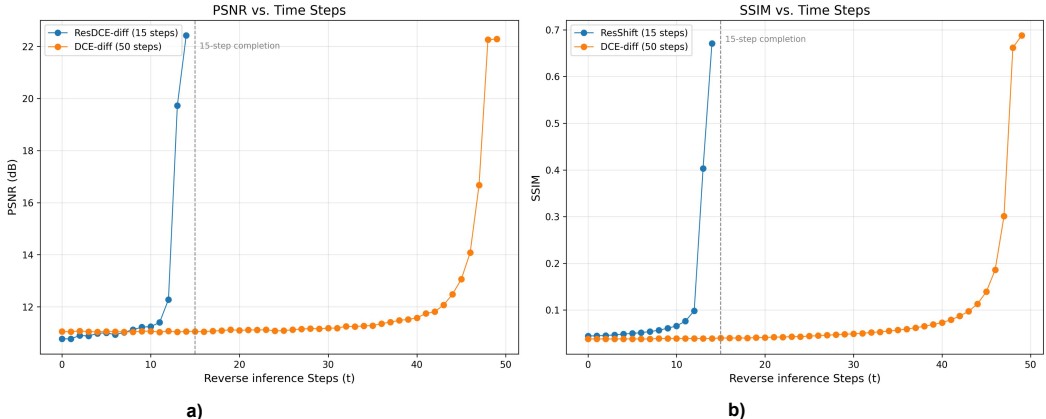

Figure 4: Comparative analysis of ResDCE-diff and DCE-diff (M et al., 2024) on reverse inference steps a) PSNR as a function of reverse steps and b) SSIM metric as a function of reverse inference steps.

Table 3: Inference time comparison between diffusion-based models.

| Method | Time per Sample (ms) | Time per Step (ms) | Efficiency Speed |
|---|---|---|---|
| ResDCE-diff | 305.92 | 20.39 | **3.56×** |
| DCE-diff | 1090.00 | 45.54 | 1.00× |

## 5.2. Impact of Performance over steps

Figure 4 a) and b) illustrates how the PSNR and SSIM evolves across the reverse denoising time steps for the ResDCE-diff (blue) and the conditional DDPM baseline DCE-diff (orange). The vertical dashed line denotes the $\approx$ 15-step completion point used for inference. ResDCE-diff achieves its final PSNR improvement within first 15 steps reaching $\approx$ 22.5 dB reflects a guided, low variance reverse trajectory. A similar trend is observed for SSIM where ResDCE-diff rapidly retains high, structural similarity within the early denoising steps, indicating improved preservation of anatomical structures and enhancement boundaries. This is due to the residual $e_0$, which encodes physiologically relevant enhancement cues correlated with T2-w and ADC. As a result, the reverse model can learn reconstructing contrast uptake patterns rather than recovering global anatomy from noise, yielding both faster convergence and higher fidelity. By contrast, DCE-diff requires many more steps ($\approx$50) before PSNR rises sharply, reconstructing early- and late- phase images from pure noise.

## 5.3. Ablation on various residual maps

To assess the importance of physiologically consistent residuals in our residual-shifting framework, we conducted a simple but informative ablation by permuting the order of

Table 4: Quantitative comparison of various residual map configurations.

| Residual configurations with same setting | EARLY-RESPONSE | | | LATE-RESPONSE | | |
|---|---|---|---|---|---|---|
| | PSNR↑ | SSIM↑ | MAE↓ | PSNR↑ | SSIM↑ | MAE↓ |
| Onset – T2-w, Early – ADC, Late – Pre-contrast (**Proposed**) | **23.39 ± 1.58** | **0.71 ± 0.08** | **0.05 ± 0.01** | **22.90 ± 1.87** | **0.68 ± 0.08** | **0.05 ± 0.01** |
| Early – T2-w, Late – ADC, Onset – Pre-contrast | 22.74 ± 1.47 | 0.66 ± 0.07 | 0.05 ± 0.01 | 22.16 ± 1.76 | 0.64 ± 0.08 | 0.06 ± 0.01 |
| Late – T2-w, Early – ADC, Onset – Pre-contrast | 22.85 ± 1.49 | 0.67 ± 0.07 | 0.05 ± 0.01 | 22.33 ± 1.74 | 0.65 ± 0.08 | 0.05 ± 0.01 |
| Onset – T2-w, Late – ADC, Early – Pre-contrast | 23.34 ± 1.49 | 0.70 ± 0.07 | 0.05 ± 0.01 | 22.94 ± 1.75 | 0.68 ± 0.08 | 0.05 ± 0.01 |
| Late – T2-w, Onset – ADC, Early – Pre-contrast | 22.20 ± 1.44 | 0.65 ± 0.07 | 0.05 ± 0.01 | 21.39 ± 1.82 | 0.62 ± 0.08 | 0.06 ± 0.01 |

Table 5: Evaluation of cross-domain generalization on unseen prostate-MRI data with the model trained on PROSTATEx dataset.

| MODEL | EARLY-RESPONSE | | | LATE-RESPONSE | | |
|---|---|---|---|---|---|---|
| | PSNR↑ | SSIM↑ | MAE↓ | PSNR↑ | SSIM↑ | MAE↓ |
| Pix2Pix | 11.41 ± 3.31 | 0.19 ± 0.07 | 0.15 | 10.03 ± 3.58 | 0.15 ± 0.06 | 0.17 |
| RegGAN | 14.96 ± 1.73 | 0.43 ± 0.08 | 0.14 | 14.54 ± 1.68 | 0.42 ± 0.08 | 0.14 |
| TSGAN | 10.74 ± 3.0 | 0.32 ± 0.11 | 0.16 | 7.62 ± 2.27 | 0.22 ± 0.07 | 0.24 |
| ResViT | 12.99 ± 1.68 | 0.34 ± 0.08 | 0.19 | 14.54 ± 2.67 | 0.42 ± 0.12 | 0.14 |
| MINet | 19.33 ± 3.66 | 0.56 ± 0.12 | 0.05 | 20.65 ± 3.44 | 0.56 ± 0.11 | 0.04 |
| Task Transformer (T2Net) | 19.97 ± 3.14 | 0.57 ± 0.11 | 0.05 | 22.47 ± 3.62 | 0.60 ± 0.11 | 0.03 |
| DCE-former | 24.91 ± 5.54 | 0.64 ± 0.20 | 0.05 | 25.64 ± 5.79 | 0.65 ± 0.13 | 0.05 |
| DCE-diff | 21.79 ± 2.47 | 0.60 ± 0.08 | 0.04 | 23.32 ± 2.58 | 0.64 ± 0.08 | 0.03 |
| ResDCE-diff | **25.69 ± 2.27** | **0.68 ± 0.08** | 0.04 | **27.29 ± 2.93** | **0.74 ± 0.08** | 0.03 |

the three synthesized DCE time points. Reordering the channels implicitly modifies the residuals $e_0 = y_0 - x_0$ and thus changes the direction of residual-shift trajectory. Table 4 shows that the original ordering of the residuals (Onset - T2, Early - ADC and Late - pre-contrast). The first row of residual configuration aligns with the natural temporal evolution of contrast enhancement consistently achieves better performance for both early- and late-response predictions than the other two configurations. This explains that the proposed residual configuration is semantically meaningful and provide the strongest guidance for the diffusion process, reinforcing the role of physiological temporal ordering in DCE-MRI synthesis.

## 5.4. Diagnostic value of synthetic images

To assess whether the synthesized DCE-MRI images preserve diagnostically relevant enhancement patterns, Figure 6 presents qualitative results comparing the early-response (DCE-2) and late-response (DCE-3) phases of DCE-Diff and ResDCE-diff. The figure 6

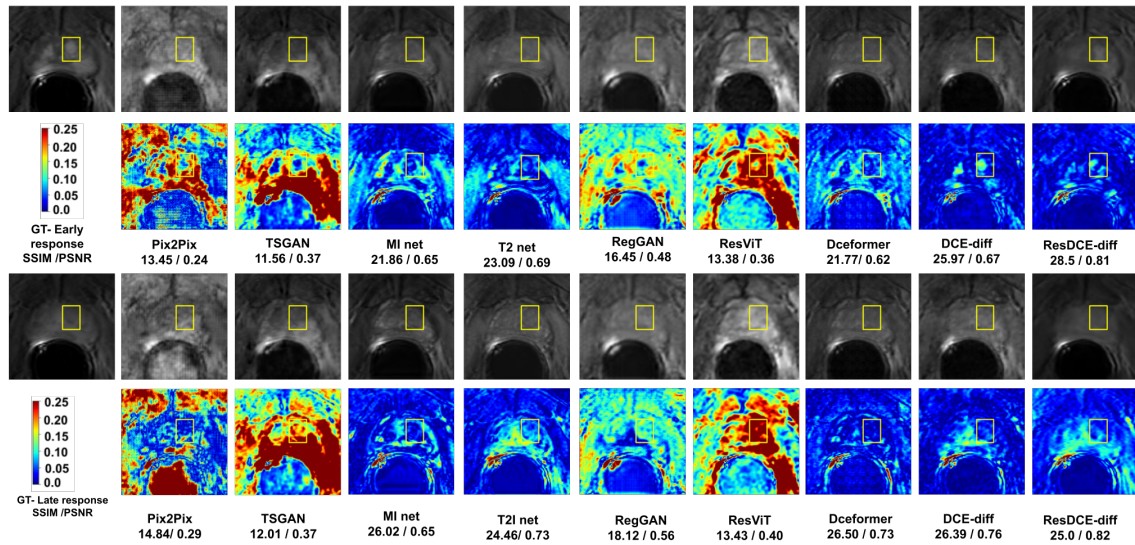

Figure 5: Qualitative results on the Prostate-MRI dataset using a model trained on the PROSTATEx dataset, demonstrating cross-dataset generalization of the proposed method. The highlighted regions indicate enhanced contrast uptake in DCE-2 phase, demonstrating that the proposed method accurately preserves the enhancement patterns.

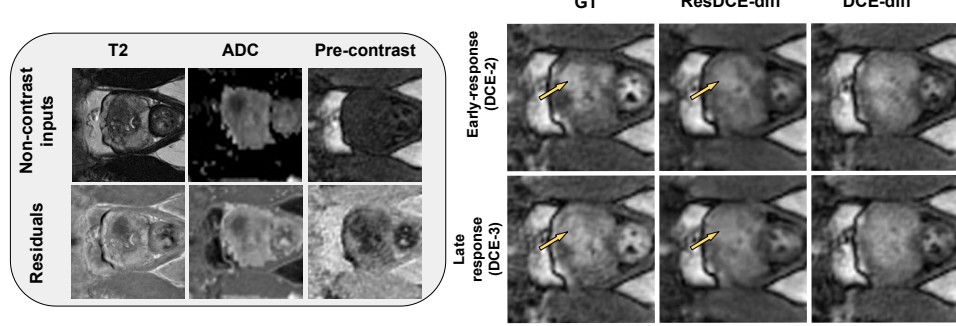

Figure 6: Diagnostic Quality experiment illustrating the perceptual consistency of reconstructions

shows that DCE-diff generates unwanted repeated patterns in the early phase response. The late phase response is noisy as compared to ResDCE-diff. Across both phases, the proposed method closely reproduces lesion conspicuity and contrast-uptake dynamics observed in the ground truth images. Overall, these qualitative results indicate that the synthesized DCE images maintain strong diagnostic integrity, accurately capturing contrast-dynamics essential for tumor characterization and treatment assessment.

### 5.5. Generalizability experiment

In order to assess the generalizability of the proposed residual diffusion framework beyond PROSTATEx dataset, we conducted an additional evaluation on another independent retrospective dataset Prostate-MRI (Choyke et al., 2016) that differs in acquisition characteristics. Prostate MRI dataset was collected using different imaging setup therefore possess domain shifts related to scanner configuration, intensity distributions, and protocol variability. The model trained on PROSTATEx dataset was tested on the Prostate-MRI dataset. These experimentation results demonstrates that the proposed model maintains stable performance and able to generalize effectively to unseen data. The quantitative results are shown in table 5 indicate that our method achieves the highest overall performance compared to the baselines. In addition, qualitative results in figure 5 further supports these findings, illustrating faithful synthesis on unseen prostate-MRI data even under domain shifts.

## 6. Conclusion

In this work, we present ResDCE-diff, a residual diffusion framework to generate early- and late-phase DCE-MRI from multi-parametric non-contrast inputs. The proposed work progressively shifts the target DCE representations using the residual maps toward their corresponding non-contrast modalities through a sequence of transition kernels enabling efficient sampling and avoiding long iterative steps. Unlike conventional diffusion based models that rely on large sampling steps our work requires only about 15 inference steps to generate contrast enhanced outputs offering a substantial gain in computational efficiency. Extensive experiments with baselines on the PROSTATEx dataset demonstrates that the proposed approach consistently outperforms state-of-the-art GAN-based and conventional diffusion-based methods with improved image quality and efficiency.

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

## Appendix A.

### A.1. Evaluation on downstream segmentation task

To investigate whether the synthesized DCE-phase images preserve task-relevant anatomical information beyond image-level similarity metrics, we conducted a downstream segmentation experiment for prostate organ segmentation i) real-DCE setting and ii) Synth-DCE setting. In real-DCE setting, all real multi-parametric inputs comprising of T2-w, ADC, DCE pre-contrast, and real DCE-phases (onset, early, and late response) stacked as 6 channel input to predict prostate organ masks. Similarly in synth-DCE setting, the real DCE-phase were replaced with the corresponding synthesized DCE-phases generated by the proposed residual diffusion model while keeping the remaining non-contrast inputs identical.

For experimentation, a 2D U-Net was employed and trained for 100 epochs with a batch size of 8 and a learning rate of $1 \times 10^{-5}$. Segmentation performance was evaluated using the mean Dice similarity coefficient and mean intersection-over-union (IoU). Quantitative and qualitative results are reported in Table 6 and in Figure 7.

As reported in Table 6 and illustrated in Figure 7, the segmentation accuracy obtained using synthesized DCE phases is comparable to that achieved using real DCE phases, indicating that the proposed method preserves task-relevant anatomical information and functional consistency. In addition to whole-image evaluation, PSNR and SSIM were computed within the prostate organ mask to focus the assessment on anatomically relevant regions as shown in Table 7. Masked evaluation reduces the influence of background tissue and acquisition-related artifacts, providing a more localized measure of fidelity for DCE signal reconstruction within the region of clinical interest.

Table 6: Organ mask segmentation performance using real and synthesized DCE images. Mean and standard deviation are reported for Dice and IoU metrics.

| Experiment Setting | Mean Dice | Mean IoU |
|---|---|---|
| Real-DCE | 0.91 ($\pm$0.08) | 0.85 ($\pm$0.10) |
| Synth-DCE | 0.90 ($\pm$0.08) | 0.84 ($\pm$0.11) |

### A.2. Uncertainty and consistency analysis

To assess inference-time uncertainty, we performed stochastic sampling by generating multiple DCE phase synthesis for an input with different random seeds. Pixel-wise variance was computed across runs for both whole-image and prostate-masked regions. The results are summarized in table 8, both the DCE-2 and DCE-3 phases exhibit low variance (i.e., $\approx 10^{-4}$), with variability within the organ-masked regions being comparable to or slightly lower than that observed over the full image. This demonstrates consistent synthesis across multiple inference runs with stable model behavior in prostate organ regions.

Along with that, figure 8 shows that distribution of pixel-wise standard deviation across multiple inference runs for DCE-2 and DCE-3 on whole image level and prostate organ level. Both the full image and prostate masked regions exhibit low variance ($\approx$ 0.009 - 0.010) demonstrating stable synthesis across diffusion sampling. The consistent distri-

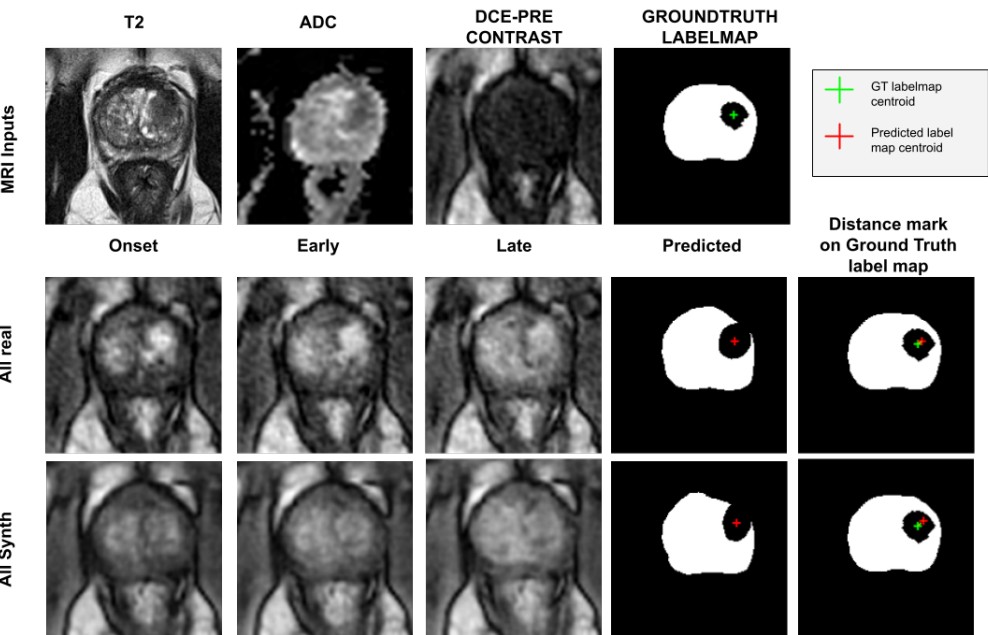

Figure 7: Downstream segmentation task on prostate organ anatomy. Top row shows the non-contrast inputs and the target prostate mask. Middle (Real-DCE setting) and bottom rows (Synth-DCE setting) shows predicted label map(red color mark) with centroids marked in red color. Column 5 (from left) shows the centroid distance between ground truth label map and predicted label maps of the corresponding experiment settings respectively.

Table 7: Quantitative evaluation of synthesized DCE images for early (DCE-2) and late (DCE-3) phases across prostate organ region.

| Phase | Region | PSNR↑ | SSIM↑ | MAE↓ |
|-------|--------|-------|-------|------|
| DCE-2 | Whole Image | $23.27 \pm 1.64$ | $0.667 \pm 0.05$ | $0.051 \pm 0.01$ |
| | Organ Region | $21.63 \pm 2.20$ | $0.570 \pm 0.06$ | $0.068 \pm 0.01$ |
| DCE-3 | Whole Image | $22.87 \pm 1.96$ | $0.650 \pm 0.06$ | $0.055 \pm 0.01$ |
| | Organ Region | $22.03 \pm 2.46$ | $0.584 \pm 0.07$ | $0.067 \pm 0.02$ |

butions across DCE phases further demonstrate the robustness of the proposed residual diffusion framework.

## A.3. Synthesis Failure analysis

Although the synthesized performance is comparable with other baselines there are cases where the model fails. To analyze failure modes and cross-modal relationships, we examined representative cases where non-contrast inputs and real DCE were inconsistent. Figure 9

Table 8: Stochastic variability of synthesized DCE-MRI images across early (DCE-2) and late (DCE-3) phases, reported as mean and standard deviation over $N = 10$ stochastic samples for full-image and organ regions.

| DCE Phase | $N$ | Full Std (mean $\pm$ std) | Organ Std (mean $\pm$ std) |
|:---:|:---:|:---:|:---:|
| DCE-2 | 10 | $0.0093 \pm 0.0010$ | $0.0090 \pm 0.0009$ |
| DCE-3 | 10 | $0.0098 \pm 0.0010$ | $0.0093 \pm 0.0008$ |

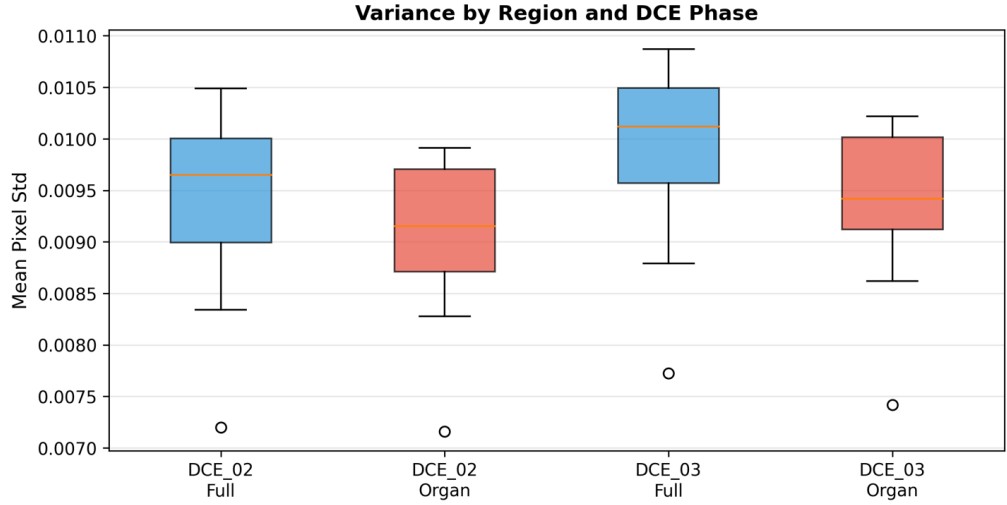

Figure 8: Box-and-whisker plot of pixel-wise variance across stochastic inference runs for synthesized DCE-02 and DCE-03 images, shown for the full image and prostate organ region.The box-and-whisker plot shows a narrow interquartile range and low variability across patients for both DCE phases, with no increase in variance within the prostate region, indicating consistent synthesis across stochastic inference runs.

illustrates two such scenarios. In (a), the T2 and ADC indicate a suspicious region, while both early and late ground truth shows no enhancement. The synthesized DCE preserves the absence of enhancement indicating that the model doesnot hallucinate contrast uptake solely based on T2 and ADC abnormalities. In contrast, (b) exhibits enhancement in ground truth DCE despite T2 and ADC appearing non-suspicious. In these cases where inputs and DCE outputs do not agree with each other, model fails to synthesize in these scenarios. These examples indicate that the model's synthesis is driven by the conditioning inputs and may fail when enhancement-related cues are not present in T2 and ADC.

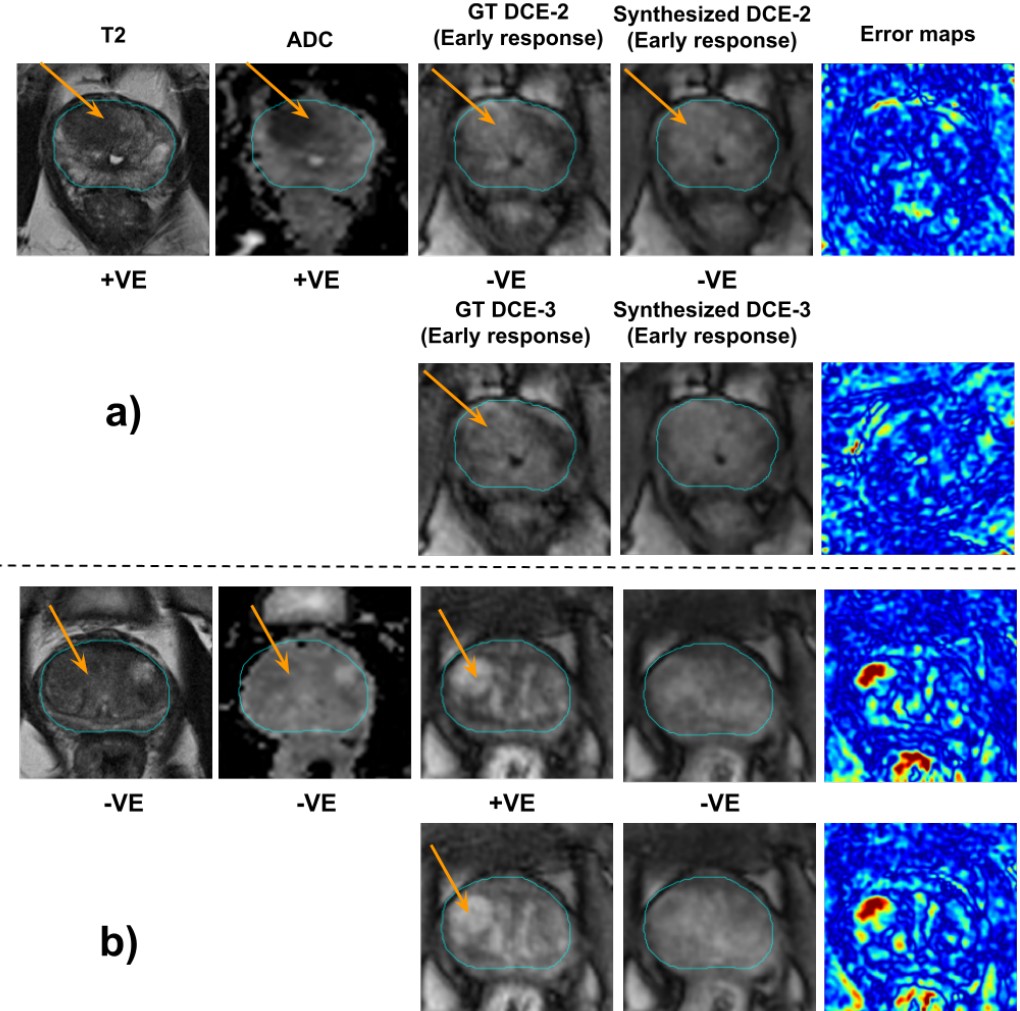

Figure 9: Representative failure cases (a) T2/ADC-positive region without corresponding enhancement in ground-truth DCE, where the synthesized DCE correctly preserves the absence of enhancement. (b) Enhancement in Groudtruth DCE is not supported by T2/ADC where the synthesized DCE underestimates the enhancement. Error map also highlights the discrepancies. In case (a), false negative (absence of lesion) in ground truth-DCE and therefore no enhancement in synthesized, whereas in case (b) inputs and DCE outputs do no agree with each other therefore model fails to synthesize.

