# OpenReview forum: "ResDCE-Diff : Dynamic contrast enhanced MRI translation in prostate cancer using residual denoising diffusion models"
_MIDL.io/2026/Conference — MIDL 2026 Poster_

### Official Review · Reviewer_jC41 · 2025-12-19

**Confidence:** 5
**Preliminary Rating:** 2

**Summary:**

The work introduces ResDCE-diff, a residual denoising diffusion model designed to synthesize early and late-phase Dynamic Contrast Enhanced (DCE) MRI sequences from non-contrast multi-modal inputs (T2-weighted, ADC, and pre-contrast T1. The authors aim to mitigate the safety concerns associated with Gadolinium-based contrast agents. Unlike standard diffusion models that initiate the reverse process from pure Gaussian noise, this method employs a residual shifting mechanism that starts from the non-contrast input distribution. The approach explicitly maps specific non-contrast modalities to specific contrast phases via residuals (e.g., ADC to early-phase)4. Experiments on the PROSTATEx dataset demonstrate that the method achieves higher PSNR and SSIM compared to GAN and standard diffusion baselines while requiring significantly fewer inference steps ~15.

**Strengths:**

The narrative arc from "Gadolinium is dangerous" to "Diffusion is slow" to "Residual Shifting is the solution" is logical and easy to follow.

Efficient sampling strategy (~15 steps) is a valuable contribution to the field of medical generative AI.

**Weaknesses:**

The premise assumes that the "residual" between an ADC map and an Early-phase DCE is a learnable, distinct feature map. This is a strong assumption, as these are fundamentally different physical properties (water diffusivity vs. vascular permeability), and the paper treats them primarily as image subtraction problems

The technical novelty is largely an application of existing work (ResShift ) to a new dataset.

The method assumes a fixed correlation between inputs (ADC/T2) and outputs (DCE phases), which ignores the biological heterogeneity of tumors (e.g., some tumors are T2-dark but not diffusion restricted)

The paper briefly mentions cross-modality synthesis limitations in PET/MRI  but fails to discuss the broader literature on "hallucination" risks in medical synthesis, which is the most critical criticism of this entire field.

Lack of clinical reader study.

Reliance on pixel-wise metrics for a problem that is inherently about functional physiology.

**Detailed Comments:**

no minor comments apart from grammar and flow check.

**Justification Of The Preliminary Rating:**

While the technical implementation of residual shifting to improve inference speed is sound and the quantitative results on PSNR/SSIM are good, the paper fails to demonstrate clinical validity. In medical imaging, particularly oncology, synthesizing contrast uptake without verifying that the tumor aggressiveness (perfusion) is accurately preserved is dangerous. The reliance purely on pixel-reconstruction metrics without a downstream clinical task or radiologist scoring makes this paper unsuitable for a high-impact venue in its current form.

**Questions To Address In The Rebuttal:**

Apart from weaknesses comments to be addressed, I have the following questions:

1. High PSNR does not equate to diagnostic accuracy. Can you provide results on a downstream task, such as tumor segmentation or classification, to demonstrate that the synthesized DCE maintains the functional characteristics of the tumors found in the PROSTATEx dataset?

2.You explicitly pair T2-w with Onset, ADC with Early-phase, and Pre-contrast with Late-phase. Given that ADC correlates with cellular density and DCE with permeability, this mapping is not strictly one-to-one. Did you attempt to learn these residual connections dynamically (e.g., using attention mechanisms) rather than hard-coding them?

3. Diffusion models allow for stochastic sampling. Have you analyzed the pixel-wise variance of the generated DCE images across multiple inference runs? High variance in tumor regions would indicate low confidence in the synthesized diagnosis.

4.Can you show examples where the model fails? specifically, are there cases where the T2/ADC shows a lesion, but the real DCE does not (or vice versa), and how does your model handle this discordance?

5.The PROSTATEx dataset is relatively homogeneous. How does your residual shifting method handle domain shifts, such as different field strengths (1.5T vs 3T) or different ADC b-value acquisitions?

---

> ### Author Response · Authors · 2026-01-25
> **We thank the reviewer for the careful evaluation of the work and providing valuable comments. The comments have significantly improved the quality of the paper**
>
> Q1)
> ----
> We thank the reviewer for emphasizing downstream evaluation. Our preliminary work focuses on the methodological feasibility of residual diffusion-based DCE synthesis with clinical validation planned as future work.  Nevertheless, we conducted segmentation-based downstream task analysis using prostate organ masks from the PROSTATEx dataset. Organ segmentation was evaluated using (i) all real inputs and real DCE phases, and (ii) real inputs with synthesized DCE phases. Mean Dice and IoU scores computed within the prostate region show comparable performance between real and synthesized settings, indicating preservation of anatomically relevant information. We further performed region-of-interest evaluation using prostate masks, where paired t-tests revealed significant differences between whole-image and organ-level metrics (p < 0.001) while maintaining stable organ-level performance across DCE phases. Additionally, lesion localization analysis using provided lesion coordinates from prostateX dataset shows close agreement between predicted and reference centroids, reflecting the increased anatomical heterogeneity and consistent synthesis. The analysis shows that the predicted and target lesion coordinates predominantly match with an error margin of 5.2in x-direction and 4.8 in y direction. The distance between them is 7 pixels. We request the reviewer to refer to the Appendix section in the revised manuscript for segmentation results.
> Manuscript changes:
> A new section added to the appendix section for explaining conducted downstream segmentation tasks on the prostate organ along with two quantitative tables Table 6 & 7 and a figure (6).
>
> Q2)
> ------
> We thank the reviewer for raising these interesting questions. We agree that T2-w and ADC are not in a strict one-to-one correspondence with respect to DCE onset and early enhancement phases respectively. Accordingly, the explicit residual subtraction between input and target modalities is therefore not intended to enforce physiological connection. We have attempted to impose a soft constraint on the diffusion model to start with a prior distribution based on the non-contrast inputs.
> We agree that dynamically learned residual connections are a potential direction to add variability in the inter-domain differences between non-contrast and DCE MRI modalities.
>
> Q3)
> ------
> We thank the reviewer for highlighting uncertainty analysis due to stochastic diffusion sampling. To address this, we conducted a pixel-wise variance analysis by generating multiple DCE phase synthesis per input using different random seeds (N=5). For both Early and late phase, the mean pixel-wise variance was on the order of 10⁻⁴, with corresponding standard deviations ≈ 0.0009.
>
> Importantly, variance within the prostate organ region was slightly lower than that of the full image, indicating stable and confident synthesis in prostate anatomy regions. These results suggest that the proposed residual diffusion model produces consistent DCE phase images across multiple inference runs without increased uncertainty in the prostate. We request the  reviewer to refer to the appendix section on this analysis. An uncertainty and consistency analysis section has been added, including quantitative results and box-and-whisker plots. Details and box-and-whisker plots are provided in the Appendix. A.2
>
> Manuscript changes
> An uncertainty and consistency analysis section has been added to the appendix A.2. Pixel-wise variance statistics for DCE-02 and DCE-03 are reported in Table 8, along with a corresponding box-and-whisker plot.
>
> Q4)
> -------
> We analyzed cases with cross-modal discordance, including lesions visible on T2/ADC without DCE enhancement and vice versa. Representative examples are included in Appendix A.3. In such cases, the model avoids hallucinating enhancement unsupported by T2/ADC and may underestimate enhancement present only in real DCE, reflecting its conditional design. This demonstrates conservative synthesis behavior and highlights limitations of relying solely on non-contrast inputs.
>
> Manuscript changes: Appendix A.3 and Figure 8 added to present failure case analysis.
>
> Q5)
> ---------
> We thank the reviewer for this observation. We acknowledge that domain shifts due to scanner vendors and acquisition protocols remain a challenge. To address this, we added a cross-dataset experiment where the model trained on ProstateX is evaluated on the Prostate-MRI dataset [1] to assess scanner-related distribution shifts. Please refer response of reviewer2

---

> ### Author Response · Authors · 2026-01-25
> **Response to the weaknesses**
>
> The premise assumes ...
> ------------------------------------
> We agree with your comment, thank you for highlighting the critical aspect.
> In general, the medical imaging community has shown interest in the context of image translation problems in various aspects like image imputation [1] and unpaired image translation [2], where the model trained on multi-contrast MR images that are not required to be captured from the same subjects. In our case, the non-contrast and DCE MRI data are paired, and the modalities correspond to a single subject. We have used prostate-x and prostate-MRI dataset both containing ADC images with lower b-values [3], which are sensitive to perfusion. Also, visually, the regions that represent lesions show correlation. Hence we have computed the residuals between modality pairs that exhibit subtle changes. Our intuition is also affirmed by our ablation study conducted with other combinations of pairs, as shown in Table 4.
> [1] CollaGAN: Collaborative GAN for Missing Image Data Imputation
> [2] A Unified Hyper-GAN Model for Unpaired Multi-contrast MR Image Translation, MICCAI 2021.
> [3] https://www.cancerimagingarchive.net/collection/prostatex/
>
>
> The technical novelty is largely ...
> ---------------------------------------------------------
> We agree with your point. We have explored the possibility of using a specific combination of residuals with subtle differences between non-contrast and DCE image pairs. Our method also gives a perspective of image imputation similar to [2, 3].
> In addition, we emphasize that the prior distribution can be arbitrary (non-Gaussian) rather than Gaussian to achieve faster inference in DCE-MRI. We note that our method demonstrates that residual shifting is applicable across diverse types of MRI data. Our aim is to show that image translation using only anatomical modalities (example - T1 to T2 MRI) can be extended to DCE-MRI
>
>
> [2] Collaborative Generative Adversarial Networks for Missing MR contrast imputation.
> [3] Handling missing values in healthcare data: A systematic review of deep learning-based imputation techniques
>
>
> The method assumes a fixed correlation between ...
> ------------------------------------------------------
> Thanks again, this is another critical aspect to consider. An extensive analysis of DCE-MRI prostate data showed that there are around 50 to 60 time points in DCE-MRI. Hence, the combination of non-contrast inputs and DCE-MRI constitutes 4D input-output paired data. Our proposed residual-shifting diffusion model is a preliminary step, using 2D time points consisting of the centre slice extracted from these volumes, to verify the feasibility of the translation task for clinical use. We plan to capture the underlying physics by extending our work using both GAN and diffusion models to 3D time points.
> Ongoing, we have a set of activities lined up, 1. to predict the pharmaco-kinetic parameters (ktrans map) using the predicted DCE-MRI timepoints as intermediate outputs. (fitting tracer kinetic models, which are computationally complex) 2. We also aim for cycle consistency, in which, given the predicted ktrans map, the DCE-MRI is predicted. [4]
> [4] Unpaired Deep Learning for Pharmacokinetic Parameter Estimation from Dynamic Contrast-Enhanced MRI
>
> The paper briefly mentions ... "hallucination" risks
> ---------------------------------------------------------
>
> The literature is predominantly available for image translation tasks in anatomical MRI modalities (Example - T1, T2, PD ) due to the availability of datasets.
> As compared to brain or knee MRI, prostate is a tougher modality for cancer imaging. Both prostate-mri and prostate-x have 26 and 345 subject respectively. The resolution of DCE-MRI is another challenge.
> Although the data are multimodal, the structural clarity of lesions is not evident in all subjects. Hallucination analysis is of critical importance in this context.
>
> The hallucination index [1] is a metric that measures the Hellinger distance between the distribution of reconstructed images and a zero-hallucination reference distribution. They have used the forward process for the zero-hallucination reference. As a quick attempt, we performed forward diffusion on the test samples and computed cosine similarity between the corresponding outputs of the forward process and in the reverse directions at time steps t = 0 to 14. The similarity values showed a minimum value of 0.732 at step 5 and higher values at the initial(t=0) and later time steps(t=15). While this is an interesting observation that might be related to the hallucination index, we believe that the correlation values are not poor.
> However, we were unable to upload this analysis within the rebuttal period. We thank the reviewer for urging us to make this analysis as it would be very helpful for extending our work.
> [1] Hallucination Index: An Image Quality Metric for Generative Reconstruction Models.
> https://papers.miccai.org/miccai-2024/370-Paper3513.html

---

### Official Review · Reviewer_pwUX · 2026-01-06

**Confidence:** 3
**Preliminary Rating:** 4
**Final Rating:** 4

**Summary:**

The work presents a novel framework designed to synthesize Dynamic Contrast Enhanced Magnetic Resonance Imaging (DCE-MRI) sequences from non-contrast multi-modal inputs, specifically T2-weighted, Apparent Diffusion Coefficient (ADC), and pre-contrast MRI images for prostate cancer. The proposed method, ResDCE-diff, uses residual denoising diffusion models to address the limitations of traditional diffusion models, which often require extensive inference steps and fail to fully utilize information present in non-contrast images. Experiments on the PROSTATEX datasets show that this approach, outperforming existing state-of-the art methods, reduces significantly the number of inference steps while improving the quality of the generated images.

**Strengths:**

The paper presents several notable strengths:
- the paper is well-written and the method carefully evaluated
- the approach and its use of residual denoising model is interesting in this context.
- results are convincing, the proposed method outperforms state-of-the art methods while reducing computational times.

**Weaknesses:**

The paper also acknowledges several weaknesses and limitations that require careful consideration:
- the approach is evaluated using a single dataset focused on prostate cancer. While the study demonstrates generalizability to real patient data within this dataset, its performance on other anatomical regions or different types of tumors remains untested. Additionally, only a single data split was employed, likely due to computational constraints. The sensitivity to residual configuration, as discussed in Section 5.3, and the observation that the original ordering of residuals yields superior results, may not generalize effectively to other datasets.
- the diagnostic value of the synthesized images lacks validation from a clinical perspective: only two samples are presented with a comparison with DCE-diff. I find the images shown on Fig 4.b difficult to interpret While quantitative assessments are provided, further evaluation by medical professionals may be necessary to confirm the reliability and applicability of these synthetic images.

**Detailed Comments:**

The paper is well written. I have identified a few minor editorial suggestions for improvement:
- Typographical Errors and Punctuation: a typo is present on page 5, where "is" should be replaced by "are" before equation (5). Punctuation is occasionally missing after equations or in figure legends/
- Attribution in Figures: Figure 1 appears to be inspired by a figure from Liu et al. It could be appropriate to credit the original source in the figure caption or legend to acknowledge the inspiration.

While the paper presents a solid approach for DCE-MRI synthesis in prostate cancer, its main limitation (the concerns about generalisability) could be more clearly stated.

**Justification Of Final Rating:**

I acknowledge that the authors’ revisions have improved the paper. However, while the amendments represent a clear enhancement, they do not fully address the concerns necessary to warrant a strong recommendation for acceptance.

**Justification Of The Preliminary Rating:**

This work presents an interesting residual denoising diffusion model designed to synthesize Dynamic Contrast Enhanced MRI (DCE-MRI) images from non-contrast multi-modal inputs for prostate cancer. The model significantly improves upon the state of the art by reducing the number of inference steps, while achieving superior image quality metrics. It uses non-constrast inputs to guide the diffusion process, ensuring physiologically meaningful and diagnostically relevant image synthesis. Despite its compelling results within the context of prostate cancer, concerns remain regarding its generalisability and robustness.

**Questions To Address In The Rebuttal:**

To improve my rating, I would appreciate more discussion on the generalisability of the approach is possible.

---

> ### Author Response · Authors · 2026-01-24
> **Addressing comments on Generalizability of proposed wotk**
>
> Q1)
> ---------
> We thank the reviewer on careful evaluation of the manuscript, raising the significant question on generalizability and pointing out minor errors.
>
> We addressed all the minor changes mentioned and provided proper citation in the revised manuscript. To further address concerns on generalizability, we additionally evaluated our model on another open source ProstateMRI dataset from the cancer imaging archive. We have now added a generalisability experiment on an independent ProstateMRI Dataset.  The learned model which is trained with the PROSTATEx dataset is tested on all patients of the ProstateMRI dataset without any changes to architecture. Quantitative results show improved PSNR/SSIM and MAE over compared methods. This shows that our method is generalizable and adaptable to another dataset that has different field strength and different modality configurations such as b values and its derived ADC comparatively with respect to baselines. The quantitative and qualitative results in the ProstateMRI dataset are added as table 5 and figure 6 in the manuscript demonstrating that the proposed method maintains stable performance under domain shift, supporting the robustness of the residual diffusion formulation.
>
> Nevertheless, we acknowledge that broader validation across multiple organs and clinical conditions remains an important direction for future work. Regarding clinical validation, we acknowledge that image-level metrics alone are insufficient for assessing medical image synthesis. Our goal is to evaluate the feasibility of residual diffusion-based DCE  synthesis rather than clinical decision making. We also acknowledge the limitation of using a single data split, which was influenced by computational constraints. We clarify that the qualitative examples are intended to demonstrate synthesis fidelity rather than clinical utility, clinical validation and expert assessment are left for future work.
>
> Q2) and Q3)
> ----------------
> Thank you for pointing this out. We addressed all the minor changes mentioned and provided proper citation in the revised manuscript.

---

> ### Author Response · Authors · 2026-01-25
> **Addressing comments on weakness**
>
> 1) We acknowledge that the current evaluation is focused on prostate DCE-MRI data. The primary aim of this work is to introduce and study residual-shift diffusion as a principled modeling strategy for multimodal-to-DCE synthesis rather than to applicability across all anatomical sites. To assess robustness under distribution shift, we train the models on the PROSTATEx dataset and evaluate them on an independent Prostate-MRI retrospective dataset for the same early and late DCE phases as well as focusing on prostate organ anatomical regions including whole-image and organ-masked settings. This evaluation provides evidence of generalization within clinically realistic variations of prostate DCE imaging.
>
> With regard to the residual configuration sensitivity discussed in section 5.3, we observe that the original residual ordering consistently yields better performance, and we believe it preserves physically meaningful progression of contrast uptake, which aligns with the temporal dynamics of DCE-MRI. Investigating alternative residual formulations and extending the framework to other anatomical regions remain important directions for future work.
>
> 2) We agree that clinical validation is an important step toward real-world adoption. In this work, our objective is to evaluate diagnostic fidelity interms of structural consistency, contrast evolution and quantitative similarity to ground-truth DCE. We acknowledge that expert radiological evaluation would further strengthen the clinical relevance of this work, and we view such validation as an important direction for future studies.

---

### Official Review · Reviewer_GEfa · 2026-01-09

**Confidence:** 5
**Preliminary Rating:** 4
**Final Rating:** 5

**Summary:**

The paper presents ResDCE-diff, a residual denoising diffusion model for synthesizing Dynamic Contrast-Enhanced MRI (DCE-MRI) images for prostate cancer assessment using non-contrast, multi-modal MRI inputs. The method is positioned as an alternative to gadolinium-based contrast agents, which are commonly used in DCE-MRI but have raised concerns related to long-term tissue deposition. Rather than initiating the diffusion process from random noise, ResDCE-diff operates on a residual formulation, iteratively refining the difference between the non-contrast input and the target DCE-MRI image. The authors evaluate the method on the ProstateX dataset, reporting quantitative results using PSNR, SSIM, and MAE, and conduct additional analyses examining the impact of diffusion steps and different residual map configurations on performance.

**Strengths:**

The main strengths of the paper lie in its use of a residual-based formulation, which effectively reduces the noise space over which the diffusion process operates, thereby decreasing the number of steps required to generate high-quality DCE-MRI images. The authors provide extensive baseline comparisons and demonstrate improved performance relative to existing methods across most evaluation metrics, with the exception of MAE in the early response setting.

**Weaknesses:**

Overall, the paper has relatively few weaknesses and presents a solid contribution. First, from an architectural perspective, it is unclear why a full 9 input–output pairing was not explored, compared to the current design that considers only 3 modality pairs. Allowing each non-contrast modality to generate any of the DCE modalities could potentially encourage a more generalized and flexible model. Second, the paper does not report inference-time statistics when comparing ResDCE-diff with DCE-diff and DCE-former. While the reduced number of diffusion steps suggests faster inference, explicitly quantifying this improvement would strengthen the evaluation.

**Detailed Comments:**

Major Points:

(1) It is unclear why a full 9 input–output pairing was not considered. Exploring all combinations between the three non-contrast modalities and the three DCE modalities could potentially lead to a more generalized and flexible model.

(2) Table 1 does not report quantitative results for the Early Onset phase. Although other baseline methods may not have produced this output, reporting the corresponding values for the proposed method would improve completeness and facilitate comparison.

(3) Inference-time statistics are not provided for ResDCE-diff, DCE-diff, and DCE-former. Including a quantitative comparison of inference times would help substantiate claims regarding the efficiency gains from the reduced number of diffusion steps.

Minor Points:

(1) In addition to PSNR, consider reporting other evaluation metrics as a function of the reverse inference steps, similar to the analysis presented in Figure 4.

(2) The caption of Figure 2 could be made more informative by explicitly describing the number of diffusion steps used (15) and briefly summarizing the process illustrated.

(3) For Figure 3, please clarify what is highlighted within the yellow box and explain why ResDCE-diff demonstrates superior performance in this region compared to the other methods.

**Justification Of Final Rating:**

I thank the authors for addressing all my concerns. I think its a strong paper with good experiments to showcase its strengths. I think the project could be significant for researchers and clinicians working with Prostate data.

**Justification Of The Preliminary Rating:**

Overall, this is a strong paper with well-designed experiments that effectively demonstrate the proposed methodology. With the suggested clarifications and additions, I would be inclined to recommend a strong accept.

**Questions To Address In The Rebuttal:**

I would like all major and minor points to be addressed. For Major Point (1), I do not expect additional experiments to be conducted; however, if there is a methodological or practical justification for not considering the full 9 input–output pairing, this rationale should be explicitly stated and discussed.

---

> ### Author Response · Authors · 2026-01-23
> **Addressing the review comments**
>
> We thank the reviewer for the constructive reviews. All the reviews were insightful and greatly improved the quality of the manuscript.
>
> Q1)
> ---------
> We thank the reviewer for this insightful question. While three non-contrast inputs and three DCE targets yield nine nominal pairings, these do not represent nine distinct outputs, as multiple inputs would redundantly predict the same DCE phase. When restricted to one-to-one assignments per DCE phase, only six meaningful modality–timepoint permutations remain, while the others introduce redundant supervision and ambiguity.
> Our design is further motivated by clinical practice, where prostate cancer assessment relies on the joint interpretation of multi-parametric MRI (T2-w, ADC, and pre-contrast). Accordingly, we focus on a multi-modal formulation rather than treating each modality as an independent generator. We evaluated multiple alternative pairings within this space, and the updated Table 3 shows that the proposed configuration consistently outperforms other permutations in terms of PSNR, SSIM, and MAE.
> Changes made in manuscript
> Table 3 has been updated with 2 more input-output paired residual configurations along with existing configurations.
>
> Q2)
> -------
> Thanks for pointing out this. We have reported the metrics of onset response of our model and updated the changes accordingly in the manuscript.
> Manuscript changes:
> Metrics for Onset-phase and its justification is added in section 5.1 in the manuscript.
>
> Q3)
> ---------
> We thank the reviewer for this suggestion. We have added an inference-time comparison for ResDCE-diff and DCE-diff under identical settings. As shown in Table 4, ResDCE-diff achieves significantly faster inference due to the reduced number of diffusion steps. DCE-former, being a GAN-based transformer, performs single-pass inference and does not rely on iterative diffusion sampling, making a direct stepwise comparison not applicable. This clarification has been added to the revised manuscript
>
> Manuscript changes
> We pointed out the sampling-step efficiency of ResDCE-diff compared with DCE-diff from Table 3 in the manuscript.
>
> Q4)
> ------
> We have added another plot for SSIM metric as a function of reverse inference steps alongside with PSNR as a function of reverse step.
> Manuscript changes:
> Added a Figure on SSIM metric analysis plot as a function of reverse inference steps in figure 4. Accordingly, captions are also modified followed by removing the below line.
>
> a) PSNR as a function of reverse steps for ResDCE-diff and DCE-diff, showing faster convergence within 15 steps. b) Diagnostic quality experiment illustrating the perceptual consistency of reconstructions. The non-contrast inputs are T2-w, ADC and pre-contrast images.
>
> Q5)
> ------
>  Yes. We have revised the caption of Figure 2 to explicitly state that 15 diffusion steps are used and briefly summarized the illustrated process.
>
> Manuscript changes:
> Figure 2 caption is elaborated and edited by replacing the below line
>
> Overview of the proposed method on DCE-MRI translation from conditional non contrast inputs. Here $x_0$ represents the onset, early- phase and late- phase images and $y_0$ represents the non-contrast multi-modal inputs such as T2-w, ADC and DCE-pre contrast. The diffusion process involves 15 timesteps for both training and inference.
>
> Q6)
> --------
> For Figure 3, please clarify what is highlighted within the yellow box and explain why ResDCE-diff demonstrates superior performance in this region compared to the other methods.
> The Region of interest has been revisited and changed, highlighting the hypo intense regions within the transverse zone of the prostate shows plausible structure with lesser residual errors compared with other methods.
>
> Manuscript changes:
> Figure 3 captions have been modified by explaining the details in the ROI

---

### Author Rebuttal · Authors · 2026-01-23

**Rebuttal:**

We sincerely thank all the reviewer for the constructive review. The comments were insightful and all reviewers have given valuable reviews that helped us to improve the quality and clarity of the manuscript significantly. Below we address each point raised by the reviewer in detail. For clarity, the reviewer’s questions are reproduced, followed by our responses. We also indicate that all the mentioned comments are revised in the manuscript and are highlighted in magenta color and removed/replaced lines are given in the below responses.

**Supporting Material:**

/attachment/a9e1a8399d4eac82142a955be62038ce9ca54e56.pdf

---

### Comment · Area_Chair_5Qem · 2026-01-31
**Final rating**

Dear reviewer, thank you for reviewing the paper! Please consider authors’ responses and update your final rating by clicking “Edit” → “Official Review” and providing the Final Rating by February 1st 2026 (23:59 AoE)!

---

> ### Comment · Area_Chair_5Qem · 2026-02-01
> **Final rating**
>
> Dear reviewer, thank you for reviewing the paper! Please consider authors’ responses and update your final rating by clicking “Edit” → “Official Review” and providing the Final Rating by February 1st 2026 (23:59 AoE)!

---

### Meta-Review · Area_Chair_5Qem · 2026-02-09

**Recommendation:** Accept (Poster)
**Confidence:** 5

**Metareview:**

Two reviewers recommend acceptance of the manuscript, while one reviewer recommends a weak reject due to concerns about clinical validity and the reader study. Based on the authors’ response, I believe the manuscript still has notable strengths that will be beneficial to the research community. Thus, I recommend acceptance of the paper.

---

### Decision · Program_Chairs · 2026-02-13

Accept (Poster)